# Active Commuting, Physical Activity, and Sedentary Behaviors in Children and Adolescents from Spain: Findings from the ANIBES Study

**DOI:** 10.3390/ijerph17020668

**Published:** 2020-01-20

**Authors:** Raquel Aparicio-Ugarriza, Juan Mielgo-Ayuso, Emma Ruiz, José Manuel Ávila, Javier Aranceta-Bartrina, Ángel Gil, Rosa M. Ortega, Lluis Serra-Majem, Gregorio Varela-Moreiras, Marcela González-Gross

**Affiliations:** 1ImFINE Research Group, Department of Health and Human Performance, Universidad Politécnica de Madrid, C/Martín Fierro 7, 28040 Madrid, Spain; apariciougarriza.raquel@gmail.com (R.A.-U.); juanfrancisco.mielgo@uva.es (J.M.-A.); 2Department of Biochemistry, Molecular Biology and Physiology, Faculty of Health Sciences, University of Valladolid, Campus de Soria, 42004 Soria, Spain; 3CIBERESP, Consortium for Biomedical Research in Epidemiology and Public Health, Carlos III Health Institute, 28029 Madrid, Spain; e.ruiz@externos.isciii.es; 4National Center for Epidemiology, Carlos III Health Institute, 28029 Madrid, Spain; 5Spanish Nutrition Foundation (FEN), C/General Álvarez de Castro 20, 1ªpta, 28010 Madrid, Spain; jmavila@fen.org.es (J.M.Á.); gvarela@ceu.es (G.V.-M.); 6CIBEROBN (Physiopathology of Obesity and Nutrition CB12/03/30038), Instituto de Salud Carlos III (ISCIII), 28029 Madrid, Spain; jaranceta@unav.es (J.A.-B.); agil@ugr.es (Á.G.); lluis.serra@ulpgc.es (L.S.-M.); 7Department of Preventive Medicine and Public Health, University of Navarra, C/Irunlarrea 1, 31009 Pamplona, Spain; 8Department of Biochemistry and Molecular Biology II, and Institute of Nutrition and Food Sciences, University of Granada, Campus de la Salud, Avda. del Conocimiento, Armilla, 18100 Granada, Spain; 9Department of Nutrition, Faculty of Pharmacy, Complutense University of Madrid, Plaza Ramón y Cajal s/n, 28040 Madrid, Spain; rortega@ucm.es; 10Research Institute of Biomedical and Health Sciences, University of Las Palmas de Gran Canaria, Preventive Medicine Service, Centro Hospitalario Universitario Insular Materno Infantil (CHUIMI), Canarian Health Service, Paseo Blas Cabrera Felipe “Físico”, 17, 35016 Las Palmas de Gran Canaria, Spain; 11Department of Pharmaceutical and Health Sciences, Faculty of Pharmacy, CEU San Pablo University, Urb. Montepríncipe, Crta. Boadilla Km. 5.3, Boadilla del Monte, 28668 Madrid, Spain

**Keywords:** walking, cycling, physical activity, sedentary behaviors, youth, ANIBES study

## Abstract

Active commuting (AC) has been proposed as a great opportunity to increase physical activity level (PA) in children and adolescents. The aim of the present study is to determine the associations between AC (walk and cycle commuting) and non-AC (motor vehicle commuting) with PA levels, and with AC and sedentarism in Spanish children and adolescents. A representative Spanish sample of 424 children and adolescents (38% females) was involved in the ANIBES (Anthropometry, Dietary Intake and Lifestyle in Spain) Study in 2013. Data on the levels of AC, non-AC, PA, and sedentarism were obtained using the International Physical Activity Questionnaire for adolescents. Stepwise backward univariate generalized linear and linear regression models were performed. In girls, walking was associated with playground PA, moderate PA, and moderate to vigorous PA (MVPA) (β *=* 0.007, *p* < 0.05; both β *=* 0.007, *p* < 0.01), respectively. In boys, walking was associated with all PA levels (*p* < 0.05); while cycling was related to moderate PA and MVPA (both β *=* 0.007, *p* < 0.05). A negative significant association was observed between AC and time spent studying without Internet use in boys (β *=* −0.184, *p* < 0.05). Commuting by walking contributes to increased daily PA in both sexes, whereas cycling was only related to moderate PA and MVPA in boys. Sedentary behaviors are not related to AC, but studying without Internet use was negatively associated with AC in boys.

## 1. Introduction

There is growing evidence that lack of physical activity (PA) and sedentarism during childhood predispose children to the development of obesity and chronic health conditions later in life [1,2]. Despite all the well-established health benefits associated with regular PA [3,4], 55.4% of Spanish children and adolescents do not meet the international recommendations for PA introduced by the World Health Organization [5] for these age groups (420 min/week of moderate and vigorous intensity PA, MVPA): these included 73.3% females and 44.5% males [6]. Moreover, the ANIBES (Anthropometry, Dietary Intake and Lifestyle in Spain) study showed that 48.4% of children and adolescents spend >2 h sitting for non-study reasons every day of the week: 49.3% during the week and 84.0% during weekends [7]. There has been a dramatic worldwide increase in the prevalence of overweight and obese children and adolescents [8]. In addition, diabetes, metabolic syndrome, and hypertension are now being identified in the youth population [9].

Motivating children to be physically active is an important public health strategy for disease prevention [3]. Active commuting (AC) to/from school provides a great opportunity to incorporate PA into the daily routine of this population [10]. Active transport to/from school, such as walking or cycling, increases PA levels [11], improves cardiovascular health, presents healthier body composition [12], reduces stress, and increases academic benefits [13]. Moreover, cycling as active transportation to/from school is positively related to weight status [14], muscular fitness [15], and cardiorespiratory fitness [16]. Furthermore, cycling seems to be more effective in improving physical fitness than other means of transport in children [11,16].

Despite all its benefits, however, AC has reduced over time in the UK [17], Australia [18], EEUU [19], Canada [20] and, dramatically, in Spain [21]. Authors agree that more research with higher quality designs and measures is needed to identify successful strategies for increasing AC to/from school [22], and for developing and implementing policy initiatives [23]. In order to create effective future interventions for promoting AC to/from school [22], it is necessary to obtain reliable and representative data on the time spent doing AC and non-AC on Spanish children and adolescents.

Therefore, our study aims to (1) describe the characteristics between PA levels split by sex, age groups, and commuting to/from school (motor vehicle, cycle, and walk); (2) examine the associations between the time spent in AC and PA levels evaluated by the International Physical Activity Questionnaire for adolescents (IPAQ-A); and (3) analyze the relationship between AC and sedentary behavior.

## 2. Material and Methods

### 2.1. Study Design

The design, protocol, and methodology of the ANIBES study have already been described in detail elsewhere [24,25]. Briefly, the ANIBES study was designed to conduct an accurate update of the PA patterns, energy expenditure, food and beverage intake, dietary behavior, and anthropometric data of the Spanish population (9–75 years, *n =* 2009).

The participants were selected from seven areas (Northeast, East, South, West, North Central, Metropolitan Barcelona, and Madrid) and the Canary Islands, in municipalities of at least 2000 inhabitants [24,25]. The selection was conducted through a stratified multistage sampling with 128 sampling points throughout Spain to guarantee more coverage and representation. No previous pre-recruitment was considered to minimize the risk of bias in responses. This paper is focused on children and adolescents (9–17 years, *n =* 424), and considers two age groups: children; (9–12 years, n *=* 213; 40.8% girls) and adolescents (13–17 years, n *=* 211; 35.1% girls). Several exclusion criteria were applied: following a therapeutic diet due to a recent surgery or any medical prescription, and suffering a transitory pathology (i.e., flu, gastroenteritis, chicken pox, etc.,) at the time of the fieldwork. Nevertheless, individuals under the following conditions were still considered eligible for inclusion: Following dietary advice for a purpose such as the prevention of diabetes; living with a diagnosed allergy and/or food intolerance; or suffering from a metabolic disease such as hyperthyroidism or hypothyroidism [24,25].

The fieldwork for the ANIBES study was conducted from mid-September 2013 to mid-November 2013 (three months), and two prior pilot studies were also scheduled (June–September, 2013). All participants were informed of the protocol and risks/benefits and all adults signed a written consent form prior to participation. In the same line, informed written consent from children and adolescents was obtained from participants and parents or guardians. The final protocol was approved by the Ethical Committee for Clinical Research of the Region of Madrid (Spain) [24,25].

### 2.2. Commuting Assessment Data

The AC and non-AC information was assessed through self-reporting using questions related to transportation from the IPAQ-A. The validation of this questionnaire can be found elsewhere [26,27]. The questionnaire was an adolescent-adapted version of the long version of the IPAQ. A pilot concurrent validation found modest correlations between the PA reported in the questionnaire and PA measured by accelerometry [27].

Children and adolescents were asked how they travelled from place to place (including places such as school, stores, theatres) as follows: (1) “During the last seven days, on how many days did you travel for at least ten uninterrupted minutes in a motor vehicle like a train, bus, car, motorbike or tram?”; (2) “How much time did you usually spend on those days travelling by motor vehicle?__hours, __minutes per day.” Both questions were repeated for cycling and walking. The recall time included all weekdays and weekends. These questions are included in the IPAQ-A [26,27]. Likewise, children and adolescents answered the question “How do you usually travel to school?” and response options were: “car, walking, cycling, bus/subway, motorcycle, other.” Participants were classified as: AC (if they used either walking or cycling commuting to and/or school) and non-AC (if they went to/from school by motor vehicle (train, bus/subway, car, motorbike or tram) [21].

### 2.3. Physical Activity Level

During a personal visit, trained researchers administered the modified IPAQ to children and adolescents according to the HELENA study [24,25]. The data collected from the IPAQ-A surveys were tallied within each PA domain to estimate the total time spent in PA related to occupational, transportation, household, and leisure activities. A detailed description of the time spent in each PA level (light, moderate, vigorous, and MVPA) has been published elsewhere [6]. Moderate PA, vigorous PA, MVPA, and PA during time spent on the playground (PA that children and adolescents perform during the break time at school) were considered in this study.

### 2.4. Sedentary Lifestyle Data

Patterns of sedentary behavior were assessed using the HELENA sedentary behavior questionnaire [28]. This questionnaire showed moderate seven-day test-retest reliability (intraclass correlation coefficients ranging from 0.36 to 0.77, and 0.71 to 0.78 for weekdays and weekends, respectively) when assessing the sedentary patterns in a sub-sample of 183 adolescents aged 13 to 18 years from the HELENA study.

Children and adolescents provided data for the hours they spent watching TV, playing computer games, playing console games, surfing the internet for non-study reasons, surfing the Internet for study, and studying (non-school time) during week days. They selected one of the following categories: (1) none; (2) less than half an hour; (3) between half an hour to an hour; (4) between one and two hours; (5) between two and three hours; (6) between three and four hours; (7) more than four hours. Participants were categorized into three groups according to the daily time sitting: <2 h; 2–4 h; >4 h.

### 2.5. Anthropometric Measurements

Weight and height were measured by trained researchers following standardized procedures. Weight was measured using a Seca 804 weighing scale (Medizinische Messsysteme und Waagen seit 1840, Hamburg, Germany; range 0.1–150 kg, precision 100 g). Height was obtained in triplicate using a Seca 206 stadiometer (Medizinische Messsysteme und Waagen seit 1840, Hamburg, Germany; range 70–205 cm, precision 1 mm). Body mass index (BMI) was calculated as weight (kg)/height (m^2^).

### 2.6. Statistical Analysis

The study sample characteristics are presented as mean, standard deviation (SD), median range, percentile 25, and percentile 75. All PA variables (AC, non-AC, different PA levels and sedentary activities) showed non-normal distribution. Participants could be using different commuting transports and all have been considered. Comparisons between groups (AC and non-AC) and PA levels were analyzed using Mann-Whitney test.

To analyze which AC outcome were associated with PA, a stepwise backward univariate generalized linear model (GLM) was used with AC (motor vehicle (min/day), cycle (min/day), and walk (min/day)) as the independent variables and the PA (playground PA (min/day), moderate PA (min/day), vigorous PA (min/day), and MVPA (min/day)) as the dependent variables. Adjusted analyses were considered in this model and GLM was also stratified by sex.

Linear regression with logarithmic transformation (AC and all sedentary activities) was calculated between cycling (min/day) and walking (min/day) as dependent variables with each sedentary activity (independent variable) stratified by sex.

All statistical analyses were run on SPSS for Windows statistical software package version 22.0 (SPSS IBM, Chicago, IL, USA). The level of statistical significance was set at a *p*-value of <0.05.

## 3. Results

Table 1 presents the descriptive characteristics of the population by age group and sex. Children and adolescents in all stages performed more than 50% of walking commuting, followed by vehicle commuting. Boys performed 17.4% and 9.2% of cycle commuting during childhood and adolescence, respectively; however, among girls, only 13.1% and 1.3% performed cycle commuting, in each age group, respectively.

Table 2 shows each PA level according to AC and non-AC. Adolescent girls who used motor commuting, also spent more time doing vigorous PA and MVPA (*p* < 0.05); however, they spent less time doing playground PA. Furthermore, both boys in both age groups who used cycle commuting, spend more time doing moderate PA than children and adolescents who did not perform cycling commuting (*p* < 0.05). Regarding walking commuting, in general, significant differences were found between girls and boys who performed walking commuting compared to those who did not perform walking commuting (*p* < 0.05).

In Table 3, girls who did walking commuting were associated with playground PA, moderate PA, and moderate to vigorous PA (MVPA) (β *=* 0.007, *p* < 0.05; β *=* 0.007, *p* < 0.01; β *=* 0.007, *p* < 0.01), respectively. Moreover, commuting by motor vehicle was only associated with vigorous PA (β *=* 0.010, *p* < 0.05). In boys, walking was associated with all PA levels (*p* < 0.05), while cycling was associated with moderate PA (β *=* 0.007, *p* < 0.05) (Table 3).

Concerning the relationship between sedentary activities and AC, there was only a significant negative association between AC and time spent studying without Internet use in boys (β *=* −0.184, *p* < 0.05) (Table 4).

## 4. Discussion

These findings suggest that there are significant associations between commuting by walking and different levels of PA in both sexes, whereas cycling has a relationship with moderate PA and MVPA in boys.

Few studies have recorded the descriptive characteristics of AC and non-AC in a representative sample of Spanish children and adolescents. Data from the THAO study corresponding to a representative sample of children aged 8 to 13 years, observed that 68% of them walked to/from school [29]. In our study, 63.4% of children (69.0% of females and 59.5% of males) and 71.6% of adolescents (71.6% of females and 71.5% of males) walked to/from school. However, only 14.1% of children (9.2% of females and 17.5% of males) and 9.0% of adolescents (1.4% of females and 13.1% of males) used cycling as an active transport. Pérez et al. [30] found that 54% of males and 57% of females aged 14 years, and 38% of males and 40% of females aged 18 years walk or use cycle as a means of transport to/from school. The results from Spain’s 2016 Report Card on Physical Activity for Children and Youth also indicate that active transport was in grade C [31]. This grade means that between 41 and 60% of the Spanish children and adolescents included in the report actively commute to/from school [31].

Our study shows that children and adolescents from both sexes who actively commute spent more time doing MVPA than those who did not do AC. Indeed, in our study, participants who used walking commuting also spent more time doing PA in all domains in both sexes (except vigorous PA in girls). In this sense, children and adolescents who usually perform PA, also carry out AC. Van Sluijs et al. [2] observed that walking to/from school in British children is associated with 5.98 (95% CI: 3.80–8.14) more minutes of MVPA per day than travelling by motor vehicle. In addition, boys were more likely to actively commute to/from school than girls in all PA domains [23,32], which is in line with our findings. Several other studies also showed similar results [33,34]. Surprisingly, our results indicate that motor vehicle commuting was associated with playground PA (boys), vigorous PA and MVPA level in girls. This may be because participants tend to overestimate the time spent doing light PA level and participants also know much better how much time they spend doing vigorous PA because of the intensity of the PA performed.

The findings of our study also suggest that children who commute by walking spend more time doing PA at all levels than adolescents who commute by walking. Regarding commuting by cycling, the time spent in all PA domains was similar between both populations in boys. In Cooper et al. [33] cycling was associated with higher overall PA only in boys. In our study, adolescent girls did not commute by cycle at all. Furthermore, Ostergaard et al. [35] observed that a school cycling promotion program did not affect school cycling behavior nor the health parameters measured in Danish children.

The perceived barriers to AC should be analyzed because they influence the decision-making of adolescents. The environment greatly affects the population in our study, and thus, the different geographical areas, weather, sidewalks, distances, and safety, for example, are all perceived barriers [36]. A recent study showed that the most recurrent barriers reported by parents of Spanish children are traffic volume and dangerous intersections, and the most common barriers reported by parents of Spanish adolescents are distance to/from school and dangerous intersections [37]. Results from the HELENA study revealed that time spent in commuting by cycling differs among European cities [22].

Children and adolescents who actively commute (cycle and walk) to/from school tend to be more physically active than non-commuters. Although active children and adolescents exhibit more AC (cycle and walk), there are some gaps in the relationship between physical fitness and AC. A recent interventional study about AC and health-related fitness observed that AC to/from school was associated with increase in rates of cycling among boys, but not of walking to/from school or health-related fitness. However, the number of children who used cycle as commuting transport was low [38].

Sedentary behaviors have increased with the emergence of motorized travel. A systematic review on children and adolescents found that active travelers spend significantly less time sitting than sedentary travelers [39]. In our study, a negative and significant association was observed between AC (cycling and walking) and time spent studying without Internet use. In this sense, participants who spent more time studying without internet use, spent less time doing AC. Martínez-Gómez et al. [13] also observed no significant relationship between watching television and AC (cycling or walking) to/from school in adolescents.

Despite the scientific evidence that shows that AC provides health benefits, public efforts are necessary to promote changes. Walking or cycling to and from school can contribute to attaining public health goals for total PA [33]. AC can be considered a simple and cost-effective method for increasing PA levels among youth populations [23]. Educational strategies are essential to promote healthy behavior among children and adolescents. However, few studies have specifically examined the effectiveness of interventions to increase active travel to/from school; the available studies have been of variable quality and produced mixed results [40].

### Strength and Limitations

The ANIBES study has several strengths, including the careful design, protocol, and methodology used and the random representative sample of the Spanish population aged 9–17 years. The validated questionnaire used to collect information on active commuting has shown good reliability and reproducibility. One limitation of this study is its cross-sectional design, which provides evidence for associations, but not for causal relationships. Although instruments to collect the information have been validated, still, there is a chance that data in its current form may overestimate the AC/PA levels and thus the relationship between them. However, a careful multistep quality control procedure was implemented to minimize the bias. Moreover, another limitation could be the high type I error rate because of hundreds of comparisons and secondary outcomes.

## 5. Conclusions

In summary, our data reveal significant associations between commuting by walking and different levels of PA in both sexes, while cycling is associated with moderate PA and MVPA in boys. Our results support the contribution of AC to overall PA and further initiatives are needed for the promotion of AC at schools. Further studies are needed in order to analyze the relationship between sedentary behavior and AC.

## Figures and Tables

**Table 1 ijerph-17-00668-t001:** Descriptive characteristics of the study sample by sex and age groups.

	Children				Adolescents			
	*n*	Boys Mean ± SD Median (P_25_–P_75_)	*n*	Girls Mean ± SD Median (P_25_–P_75_)	*n*	Boys Mean ± SD Median (P_25_–P_75_)	*n*	Girls Mean ± SD Median (P_25_–P_75_)
**Body mass (kg)**	126	43.0 ± 9.8	87	43.3 ± 10.3	137	64.0 ± 13.7	74	57.4 ± 10.4
**Height (cm)**	126	147.0 ± 9.6	87	147.3 ± 9.52	137	171.1 ± 9.2	74	161.5 ± 7.0
**BMI (kg/m^2^)**	126	19.7 ± 3.4	87	19.7 ± 3.2	137	21.8 ± 3.4	74	21.9 ± 3.5
**Geographical Distribution (%)**								
**Center**	22	17.5	18	20.7	25	18.2	18	24.3
**Atlantic**	19	15.1	20	23.0	31	22.6	17	23.0
**Mediterranean**	42	33.3	26	29.9	52	38.0	15	20.3
**South**	43	34.1	23	26.4	29	21.2	24	32.4
**Parents Income (%)**								
**<1000€**	14	15.7	19	31.7	20	21.1	10	19.6
**1000–2000€**	60	67.4	31	51.7	57	60.0	31	60.8
**>2000€**	15	16.9	10	16.7	18	18.9	10	19.6
**Commuting Model (%)**								
**Used walk commuting**	75	59.2	60	69.0	98	71.5	53	71.6
**Used cycle commuting**	22	17.4	8	9.2	18	13.1	1	1.3
**Used motor vehicle commuting**	47	37.3	29	33.3	60	43.8	34	45.9

BMI, body mass index.

**Table 2 ijerph-17-00668-t002:** Descriptive characteristics between PA levels divided by sex, age groups and commuting to/from school (motor vehicle, cycle, and walk).

Yes/No Motor Vehicle Commuting.												
	Boys						Girls					
	9–12 Years		13–17 Years		Total		9–12 Years		13–17 Years		Total	
	Median (P_25_–P_75_)		Median (P_25_–P_75_)		Median (P_25_–P_75_)		Median (P_25_–P_75_)		Median (P_25_–P_75_)		Median (P_25_–P_75_)	
	Yes	No	Yes	No	Yes	No	Yes	No	Yes	No	Yes	No
**Moderate PA** **(min/day)**	42.9 (15.7–120.0)	40.7 (0–68.6)	21.4 (0–65.4)	12.9 (0–51.4)	30.0 (0–75.7)	23.6 (0–60.0)	0 (0–51.4) ^†^	28.6 (17.4–54.3) ^†^	11.4 (0–34.3)	5.4 (0–22.1)	0 (0–42.9)	20.4 (0–40.7)
**Vigorous PA** **(min/day)**	47.1 (25.7–77.1)	34.3 (11.4–51.4)	34.3 (0–56.4)	20.0 (0–64.3)	34.3 (0–68.6)	28.9 (0–60.0)	17.4 (0–34.3)	5.4 (0–25.7)	8.6 (0–34.3) ^†^	0 (0–13.6) ^†^	17.1 (0–34.3)	0 (0–22.9)
**MVPA** **(min/day)**	77.9 (44.3–201.4)	70.1 (34.3–120.0)	55.7 (15.0–125.7)	55.7 (17.1–111.4)	71.4 (34.3–150.0)	62.9 (21.4–116.8)	34.3 (0–91.4)	44.6 (18.6–100.7)	32.1 (0–54.3) ^†^	13.9 (0–35.4) ^†^	34.3 (0–74.3)	25.7 (0–60.0)
**Playground PA** **(min/day)**	64.0 (30.0–116.0)	59.0 (22.0–120.0)	37.0 (15.0–98.0)	18.0 (0–50.0)	50.0 (18.0–105.0) *	31.0 (12.0–82.0) *	30.0 (15.0–58.0)	41.0 (15.0–64.0)	22.0 (0–51.0) ^†^	5.0 (0–25.5) ^†^	30.0 (15.0–58.0)	24.5 (0–60.0)
**Yes/No Cycling Commuting.**												
**Moderate** **PA (min/day)**	53.6 (47.1–107.1)	31.4 (0–72.1)	64.3 (20.0–145.5)	15.0 (0–51.4)	56.1 (25.7–128.6) *	21.4 (0–60.0) *	50.4 (32.1–95.0)	21.4 (0–51.4)	-	10.7 (0–30.0)	40.7 (30.0–74.3) *	17.1 (0–38.9) *
**Vigorous** **PA (min/day)**	35.0 (25.7–60.0)	39.3 (15.0–70.7)	25.7 (0–77.1)	25.7 (0–60.0)	34.3 (13.9–62.9)	34.3 (0–64.3)	13.9 (0–29.6)	10.7 (0–25.7)	-	0 (0–19.3)	17.1 (0–34.3)	0 (0–25.0)
**MVPA (min/day)**	94.6 (57.9–167.1)	69.3 (34.4–135.0)	141.4 (21.4–180.0) ^†^	51.4 (12.7–111.4) ^†^	101.4 (51.4–171.4) *	62.1 (24.1–120.0) *	64.3 (40.7–116.1)	34.3 (10.7–100.7)	-	17.1 (0–42.9)	60.0 (34.3–91.4)	25.7 (0–57.9)
**Playground PA (min/day)**	60.0 (35.0–176.0)	56.5 (23.0–107.0)	36.0 (0–154.0)	24.0 (0–69.0)	60.0 (24.5–164.5) *	36.0 (15.0–90.0) *	75.0 (13.5–130.0)	32.0 (15.0–60.0)	-	15.0 (0–32.0)	54.0(0–120.0)	25.0 (5.0–176.0)
**Yes/No Walk Commuting.**												
**Moderate** **PA (min/day)**	51.4 (18.6–85.7) ^†^	25.7 (0–57.1) ^†^	2.6 (0–68.6) ^†^	12.9 (0–34.3) ^†^	38.6 (0–77.1) *	16.4 (0–51.4) *	28.6 (0–74.6) ^†^	12.9 (0–35.0) ^†^	15.0 (0–31.4) ^†^	0 (0–11.4) ^†^	20.7 (0–51.4) *	14.8 ± 23.4 0 (0–22.1) *
**Vigorous** **PA (min/day)**	43.6 (17.1–77.1)	30.7 (12.9–51.4)	33.2 (0–61.4)	0 (0–51.4)	34.2 (11.4–72.9) *	25.7 (0–51.4) *	11.8 (0–32.1)	10.7 (0–25.7)	0 (0–22.9)	0 (0–0)	0 (0–25.7)	0 (0–19.3)
**MVPA (min/day)**	85.7 (57.1–167.1) ^†^	51.4 (25.7–117.9) ^†^	62.1 (20.0–128.6) ^†^	34.3 (10.0–90.0) ^†^	80.0 (38.6–145.7) *	47.1 (17.1–102.9) *	45.7 (17.1–120.0) ^†^	21.4 (0–51.4) ^†^	30.0 (0–43.9) ^†^	0 (0–22.9) ^†^	34.3 (10.7–89.3) *	17.1 (0–44.6) *
**Play–ground PA (min/day)**	78.0 (36.0–156.0) ^†^	35.0 (20.0–74.0) ^†^	36.0 (12.0–86.0) ^†^	15.0 (0–20.0) ^†^	60.0 (20.0–120.0) *	20.0 (12.0–54.0) *	42.0 (17.5–80.5)	30.0 (15.0–60.0)	20.0 (0–36.0)	0 (0–30.0)	30.0 (12.0–62.0) *	24.0 (0–30.0) *

AC: active commuting; PA: physical activity; MVPA, moderate to vigorous intensity of physical activity. Mann-Whitney test was used for non-normally distributed variables. * Comparisons between Yes/no AC and non-AC and within each sex (*p* < 0.05). ^†^ Comparisons between Yes/no AC and non-AC and within each sex and each age group (*p* < 0.05).

**Table 3 ijerph-17-00668-t003:** Association between AC and non-AC with PA levels divided by sex.

Playground PA (min/day)			
	**Adjusted Analysis**		
	**β**	**R^2^**	***p***
**Boys**			
Motor vehicle (min/day)	0.005	0.191	0.003
Cycle (min/day)	-		-
Walk (min/day)	0.010		0.000
**Girls**			
Motor vehicle (min/day)	-	0.040	-
Cycle (min/day)	-		-
Walk (min/day)	0.007		0.028
**Moderate PA (min/day)**			
**Boys**			
Motor vehicle (min/day)	-	0.118	-
Cycle (min/day)	0.007		0.023
Walk (min/day)	0.006		0.001
**Girls**			
Motor vehicle (min/day)	-	0.147	-
Cycle (min/day)	-		-
Walk (min/day)	0.007		0.004
**Vigorous PA (min/day)**			
**Boys**			
Motor vehicle (min/day)	-	0.041	-
Cycle (min/day)	-		-
Walk (min/day)	0.005		0.005
**Girls**			
Motor vehicle (min/day)	-	0.041	-
Cycle (min/day)	-		-
Walk (min/day)	0.005		0.005
**Moderate to Vigorous PA (min/day)**			
**Boys**			
Motor vehicle (min/day)	-	0.073	-
Cycle (min/day)	-		-
Walk (min/day)	0.007		0.000
**Girls**			
Motor vehicle (min/day)	0.010	0.158	0.026
Cycle (min/day)	-		-
Walk (min/day)	0.007		0.006

PA, physical activity. Data are reported as the standard regression coefficients point estimate with the associated *p*-value. Adjusted model by age groups using stepwise backward test. Bold numbers are used to highlight a significant result. Level of significance *p* < 0.05.

**Table 4 ijerph-17-00668-t004:** Associations between AC and each sedentary activity during the week by divided by sex.

	AC (Cycle and Walk) (min/day)		
Spend Time in Sedentary Activities (<2 h; 2.4 h; >4 h)	β	R^2^	*p*
**Boys**			
TV viewing	0.044	0.002	0.558
Console games	0.062	0.004	0.410
Computer games	0.138	0.019	0.068
Internet use for non-study reason	−0.062	0.004	0.413
Internet for academic reason	−0.015	0.001	0.845
Study without internet use	−0.184	0.034	**0.013**
**Girls**			
TV viewing	0.021	0.006	0.829
Console games	−0.044	0.008	0.644
Computer games	−0.009	0.006	0.925
Internet use for non-study reason	0.000	0.006	0.997
Internet for academic reason	−0.075	0.012	0.429
Study without internet use	−0.028	0.007	0.768

Data are reported as the standard regression coefficients point estimate with the associated *p*-value. Analyses are adjusted by age groups. Bold numbers are used to highlight a significant result. Level of significance *p* < 0.05.

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
