# Peer review of "Active Commuting, Physical Activity, and Sedentary Behaviors in Children and Adolescents from Spain: Findings from the ANIBES Study"

_ijerph, 2020, doi:10.3390/ijerph17020668_

Round 1
Reviewer 1 Report
This study deals with youth active commuting and provides valuable information.
I think that the proper design of the research is adopted, and it compensates the weaknesses of cross-sectional research and self-report.
However, I want you to correct the arrangement of Table 2 and Table 3. Because they are spread over two pages, it is difficult to see.
Author Response
Dear Reviewer,
We appreciate the time you devoted to reading our manuscript and helping us to craft an improved version. We are pleased to clarify your concerns and to add your comments to the manuscript which we believe will improve the impact and quality of our work in order to publish it in International Journal of Environmental Research and Public Health. Please find below our response to each of your observations. We have made a concerted attempt to systematically address the specific concerns raised for this revision and we have highlighted all the alterations to this revision within the manuscript in yellow for your convenience.
REVIEWER 1
REVIEWER: This study deals with youth active commuting and provides valuable information. I think that the proper design of the research is adopted, and it compensates the weaknesses of cross-sectional research and self-report.
AUTHORS: We greatly appreciate your comment. Thank you.
REVIEWER: However, I want you to correct the arrangement of Table 2 and Table 3. Because they are spread over two pages, it is difficult to see.
AUTHORS: Thank you for your recommendation. The authors have proceeded to adjust Table 2 and 3 on a single sheet.

Reviewer 2 Report
The study is well written, despite the methodological limitations you have highlighted.
However, in my opinion, a study with data collected 6 years ago cannot be published. This study was to be sent to a journal in 2014, but not at the end of 2019.
Author Response
Dear Reviewer,
We appreciate the time you devoted to reading our manuscript and helping us to craft an improved version in order to publish it in International Journal of Environmental Research and Public Health. Please find below our response to your observation.
REVIEWER 2
The study is well written, despite the methodological limitations you have highlighted.
REVIEWER: However, in my opinion, a study with data collected 6 years ago cannot be published. This study was to be sent to a journal in 2014, but not at the end of 2019.
AUTHORS: Thank you for your comment. Although the data collected 5-6 years ago could be outdated with respect to the present, we have analyzed associations, which are valid data over time. Additionally, the results obtained were discussed with current references. Therefore, the results do not invalidate the conclusions obtained for that sample at that time.
We would also like to add that data from the ANIBES study are the latest available data on a representative sample for Spain. As we clearly indicate date of data collection, this data can contribute to analyze trends over time. Unfortunately, trends regarding physical activity and sedentarism are scarce for Spain in the past.

Reviewer 3 Report
The study aims to determine the association of active commuting with PA levels and sedentary behaviour in children and adolescents from Spain. The sample consists of ~400 participants and is representative of the Spanish population ages 9-17. That’s a major strength since the results should be easily extrapolated to the overall target population. Also, some of the authors have been involved in other multicentric studies, so I assume they have enough experience to conduct this kind of study. The subject is of interest because active commuting is available to anyone at no or relatively low cost, also at low risk, and so it might be a good starting point to enhance physical activity and active living.
General comments
I suggest a thorough language editing hopefully by an English native speaker.
Specific comments
The study is highly reliant on self-report data of active commuting and physical activity. Although instrument to collect the information have been validated, still, there is a chance that data in its current form may overestimate the AC/PA levels and thus the relationship between them. There is nothing wrong with using self-report PA, overall or subcategories, but this is a major limitation of the study, and thus, it must be reckoned as such by the authors.
What do you mean by ‘playground PA’?
If all variables are non-normally distributed, why were describe using mean and SD?
Because the sample consists of children ages 9-17, I’d rather use BMI-for-age-and-sex instead of raw BMI. Also, would you consider providing the prevalence of overweight and obesity in the sample? I recommend using the 2007 WHO standard by De Onis and collaborators to define weight status categories.
Once you have the weight status of participants, is it possible to use it a control variable, in the same way sex was used as control variable?
Table 4 should specify which group is the reference category (eg. participants having <2 h/day). Also, because each variable was categorized using 3 groups, there should be two coefficients along with the intercept, which is the coefficient for the reference category. The authors may wanna see the following: https://www.southampton.ac.uk/passs/confidence_in_the_police/multivariate_analysis/simple_linear_regression_several_categories.page
Author Response
Dear Reviewer,
We appreciate the time you devoted to reading our manuscript and helping us to craft an improved version. We are pleased to clarify your concerns and to add your comments to the manuscript which we believe will improve the impact and quality of our work in order to publish it in the International Journal of Environmental Research and Public Health. Please find below our response to each of your observations. We have made a concerted attempt to systematically address the specific concerns raised for this revision and we have highlighted all the alterations to this revision within the manuscript in yellow for your convenience.
REVIEWER 3
REVIEWER: The study aims to determine the association of active commuting with PA levels and sedentary behaviour in children and adolescents from Spain. The sample consists of ~400 participants and is representative of the Spanish population ages 9-17. That’s a major strength since the results should be easily extrapolated to the overall target population. Also, some of the authors have been involved in other multicentric studies, so I assume they have enough experience to conduct this kind of study. The subject is of interest because active commuting is available to anyone at no or relatively low cost, also at low risk, and so it might be a good starting point to enhance physical activity and active living.
AUTHORS: We greatly appreciate your comment. Thank you.
General comments
REVIEWER: I suggest a thorough language editing hopefully by an English native speaker.
AUTHORS: Thanks so much for your help. The manuscript has undergone English language editing by an English native speaker. The text has been checked for correct use of grammar and common technical terms, and edited to a level suitable for reporting research in a scholarly journal.
Specific comments
REVIEWER: The study is highly reliant on self-report data of active commuting and physical activity. Although instrument to collect the information have been validated, still, there is a chance that data in its current form may overestimate the AC/PA levels and thus the relationship between them. There is nothing wrong with using self-report PA, overall or subcategories, but this is a major limitation of the study, and thus, it must be reckoned as such by the authors.
AUTHORS: Thank you for your observation. In order to clarify this concern, we have added the following text in the Strength and limitations section.
Although instruments to collect the information have been validated, still, there is a chance that data in its current form may overestimate the AC / PA levels and thus the relationship between them. However, a careful multistep quality control procedure was implemented to minimize bias.
REVIEWER: What do you mean by ‘playground PA’?
AUTHORS: Thank you for your interest. Playground is the physical activity that children and adolescents perform during the break time at school. In this line, we have included in the Physical activity level section the following sentence: “Moderate PA, vigorous PA, MVPA and PA during time spent on the playground (PA that children and adolescents perform during the break time at school) were considered in this study.”
REVIEWER: If all variables are non-normally distributed, why were describe using mean and SD?
AUTHORS: Thanks for your comment. The body mass, height and BMI were described using means and standard deviation because they are normally distributed. The data that presented a non-normally distribution were presented as Median (P25- P75). In order to avoid misunderstandings, we have indicated in the Statistical analysis section that all the variables related to PA presented non-normally distribution: “All PA variables (AC, non-AC, different PA levels and sedentary activities) showed non-normal distribution.”
REVIEWER: Because the sample consists of children ages 9-17, I’d rather use BMI-for-age-and-sex instead of raw BMI. Also, would you consider providing the prevalence of overweight and obesity in the sample? I recommend using the 2007 WHO standard by De Onis and collaborators to define weight status categories.
AUTHORS: Thank you for your interest. Data related to the prevalence of overweight and obesity were presented in another article of the ANIBES study [1] using age- and sex-specific cutoff values according to the criteria of Cole et al. [2], which have been adopted by the International Obesity Task Force.
[1] Pérez-Rodrigo, C., Gil, Á., González-Gross, M., Ortega, R., Serra-Majem, L., Varela-Moreiras, G., & Aranceta-Bartrina, J. Clustering of dietary patterns, lifestyles, and overweight among Spanish children and adolescents in the ANIBES study. Nutrients 2016, 8(1), 11.
[2] Cole, T.J.; Bellizzi, M.C.; Flegal, K.M.; Dietz, W.H. Establishing a standard definition for child overweight and obesity worldwide: International survey. BMJ 2000, 320, 1240–1243.
REVIEWER: Once you have the weight status of participants, is it possible to use it a control variable, in the same way sex was used as control variable?
AUTHORS: Thank you for this nice detail. The authors performed the statistical analysis adjusting for other variables such as weight and BMI, but as there were no differences, so it was decided to only adjust for sex and age.
REVIEWER: Table 4 should specify which group is the reference category (eg. participants having <2 h/day). Also, because each variable was categorized using 3 groups, there should be two coefficients along with the intercept, which is the coefficient for the reference category. The authors may wanna see the following: https://www.southampton.ac.uk/passs/confidence_in_the_police/multivariate_analysis/simple_linear_regression_several_categories.page
AUTHORS: Thank you for your observation. To meet the objective of quantifying the dependence between Active Commuting and sedentary variables, it was not considered necessary to conduct a GLM to identify change by category. In fact, a ratio of overall change was sought, so it was considered sufficient to carry out a simple linear model. In addition, virtually none of the independent variables were related and it was not considered necessary to move to a univariate model of variance

Round 2
Reviewer 1 Report
I think the two tables have been improved and made easier to read.
Reviewer 2 Report
Thanks for the clarifications and the answers to my comment.
Reviewer 3 Report
All my concerns were addressed in the rebuttal. Congrats to the authors and I hope you keep working on this key topic.